

# An attention bias test to assess anxiety states in laying hens

Dana L.M. Campbell[1,2], Peta S. Taylor[2], Carlos E. Hernandez[1,2,3], Mairi Stewart[4,5], Sue Belson[1] and Caroline Lee[1,2]

[1] Agriculture and Food, Commonwealth Scientific and Industrial Research Organisation (CSIRO), Armidale, NSW, Australia
[2] School of Environmental and Rural Science, University of New England, Armidale, NSW, Australia
[3] Department of Animal Nutrition and Management, Swedish University of Agricultural Sciences, Uppsala, Sweden
[4] AgResearch, Ruakura Research Centre, Hamilton, New Zealand
[5] Current affiliation: Ministry for Primary Industries, Wellington, New Zealand

Corresponding author
Dana L.M. Campbell,
Dana.Campbell@csiro.au

## ABSTRACT

Fear is a response to a known threat, anxiety is a response to a perceived threat. Both of these affective states can be detrimental to animal welfare in modern housing environments. In comparison to the well-validated tests for assessing fear in laying hens, tests for measuring anxiety are less developed. Perception of a threat can result in an attention bias that may indicate anxious affective states in individual hens following playback of an alarm call. In Experiment 1, an attention bias test was applied to hens that differed in their range access to show that hens that never ranged were more vigilant (stretching of the neck and looking around: $P < 0.001$) and slower to feed following the second alarm call playback ($P = 0.01$) compared with hens that ranged daily. All hens showed a reduction in comb temperature following the first alarm call ($P < 0.001$). In Experiment 2, an open field test was used to determine an effective dose of 2 mg/kg for the anxiogenic drug *meta*-Chlorophenylpiperazine (*m*-CPP) in adult laying hens. Hens dosed with 2 mg/kg showed reduced locomotion compared with a saline solution ($P < 0.05$). In Experiment 3, 2 mg/kg *m*-CPP or saline was administered to adult hens previously habituated to the open field arena to pharmacologically validate an attention bias test as a measure of anxiety. Hens dosed with *m*-CPP were slower to feed ($P = 0.02$) and faster to vocalize following a second alarm call playback ($P = 0.03$) but these hens did not exhibit the same vigilance behavior as documented in Experiment 1. The *m*-CPP hens also spent more time stepping and vocalizing (both $P < 0.001$) than the saline hens. An attention bias test could be used to assess anxiety. However, behavioral responses of hens may vary depending on their age or test environment familiarity, thus further refinement of the test is required. In these tests, 2 mg/kg of *m*-CPP resulted in motionless behavior when the environment was novel, but more movement and vocalizing when the environment was familiar. The extreme behavioral phenotypes exhibited by individually-tested birds may both be indicators of negative states.

## INTRODUCTION

Fear and anxiety in animals are emotional responses to a threat that stimulate adaptive behaviors such as escape or defense (*Steimer, 2002*). While fearful and anxious states are entangled in some respects and have overlap in the underlying neurological mechanisms (*Davis, 1992*), they can be distinguished as responses to known (fear) and unknown (anxiety) dangers (*Steimer, 2002*). Fear responses are adaptive for self-preservation but an excessive or inappropriate fear response can result in stress, injury, pain, or development of abnormal behavior and is thus considered a negative welfare state (*Barnett et al., 1994*; *De Haas et al., 2014a*; *Fraise & Cockrem, 2006*; *Jones, 1996*; *Uitdehaag et al., 2008*). The emotional state of anxiety may also result in similar negative impacts (*Ohl, Arndt & Van Der Staay, 2008*) but the assessment of this state in laying hens is less defined than measurement of fear (see *Forkman et al., 2007* for a critical review of fear tests).

In laying hens, an open field test has previously been used as a measure of anxiety (*Nordquist et al., 2011*) although it is most commonly categorized as fear assessment (*Gallup & Suarez, 1980*; *Suarez & Gallup, 1982*). Typically an open field test in chickens is interpreted as representing a balance between anti-predatory behavior manifested as freezing, and a desire for social contact, manifested as vocalizations (*Gallup & Suarez, 1980*; *Vallortigara, 1988*; *Vallortigara & Zanforlin, 1988*). A chicken's desire for social reinstatement when placed into a novel and isolated environment has also been used as an indication of anxiety (*De Haas et al., 2014b*). Attention bias toward a threat is a further potential measure of anxiety (reviewed in *Crump, Arnott & Bethell, 2018*). When presented with a threat, anxious individuals will show increased or biased attention toward that threat including increased vigilance behavior and a reduced willingness to feed (*Bethell et al., 2012*; *Bradley, Mogg & Lee, 1997*; *Eilam, Izhar & Mort, 2011*; *Lee et al., 2016*, *2018*; *Monk et al., 2018a*, *2018b*). This attention and vigilance behavior can be assessed as an indicator of the anxious state of the animal. *Brilot & Bateson (2012)* first demonstrated the importance of attention bias toward threats as a measure of welfare in starlings. Birds that were denied access to a water bath, which is important for maintaining feather condition, were more vigilant toward conspecific alarm calls than those birds that were provided a water bath. This study used threat perception to indicate that birds denied bathing opportunities were in a more anxious state and that water baths were important for starling welfare.

Attention bias toward threats has recently been pharmacologically validated as a measure of anxiety in sheep (*Lee et al., 2016*; *Monk et al., 2018a*, *2018b*) and cattle (*Lee et al., 2018*) using the anxiogenic drug *meta*-Chlorophenylpiperazine (*m*-CPP). In the recent validation tests, sheep or cattle that were briefly presented a dog while dosed with *m*-CPP showed increased vigilance and attention toward the previous location of the dog relative to control animals (*Lee et al., 2016*, *2018*; *Monk et al., 2018a*, *2018b*). Internal temperature loggers were also used in these studies to demonstrate stress-induced hyperthermia as a measurable physiological temperature response to the threat while under pharmacological influence (*Lee et al., 2018*; *Monk et al., 2018a*, *2018b*). In laying

hens, observations of vigilance have been made in individual or group settings to assess the fear or stress status of birds in response to different environmental conditions (e.g., group size during perching: *Newberry, Estevez & Keeling, 2001*; perch height: *Brendler, Kipper & Schrader, 2014*; presence of cockerels: *Odén et al., 2005*; a novel arena: *Rutherford et al., 2003*; presence of predator odors: *Zidar & Løvlie, 2012*), including differences in vigilance in response to an acoustical disturbance (*Brendler, Kipper & Schrader, 2014*). *Favati, Leimer & Løvlie (2014)* also measured vigilance following playback of an alarm call and correlated this with dominance in male domestic fowl. *Moriarty (1995)* gave an anxiogenic β-carboline to layer chicks and found longer durations of tonic immobility and reduced movement and vocalizations in an open field test compared to control chicks which was interpreted as increased fear and anxiety. However, to the best of the authors' knowledge, there is no research on pharmacologically validating an individual-bird test of attention bias following perception of a threat (alarm call) in layers that could be used as a standard welfare assessment test for domestic hens. The drug $m$-CPP was selected for pharmacological manipulation of anxiety based on the previous success with cattle (*Lee et al., 2018*) and sheep (*Lee et al., 2016*; *Monk et al., 2018a*, *2018b*). Physiologically, peripheral temperatures on a hen's head can also be measured as an indicator of a stress-induced hyperthermic response where elevated core temperature correspondingly results in a decrease in peripheral temperatures (*Edgar et al., 2011*, *2013*; *Herborn et al., 2015*).

Radio-frequency identification (RFID) technology has demonstrated that individual free-range laying hens, both in commercial and experimental settings, will vary greatly in their range use (*Campbell et al., 2017*; *Gebhardt-Henrich, Toscano & Fröhlich, 2014*, *Larsen et al., 2018*). Although the majority of hens will visit the range on most days, there are small percentages of hens that will rarely or never go outside (low ranging). External health assessments have, to date, not identified specific illness or injury in these low-ranging birds, suggesting they may vary in behavioral traits which impacts their choice to not go outdoors (*Campbell et al., 2016*; *Larsen et al., 2018*). Correlations of range use with individual responses in a compilation of tests (tonic immobility, open field tests, manual restraint tests, novel object tests) have shown that indoor-preferring hens present higher levels of fear and greater stress responses compared to birds that regularly use the range (*Campbell et al., 2016*; *Hartcher et al., 2016*; *Hernandez et al., 2014*; *Larsen et al., 2018*), although not consistently across every behavioral test in every study. Within a free-range system, the indoor barn is a more controlled and predictable environment than the outdoor range where birds can be exposed to variable and changing conditions, including predators. Indoor birds may choose to stay indoors because they are also more anxious than hens that regularly go outside. Previous research on personality assessment in domestic fowl has confirmed repeatable personality differences in fowl that are also correlated with other behavioral profiles such as coping style or some aspects of learning (*Zidar et al., 2017*, *2018*).

The objectives of the current study were threefold, presented as they occurred chronologically. The first objective was to use an attention bias test with an alarm call playback to assess differences in behavioral and peripheral temperature responses between free-range hens that had used the range every day (outdoor) compared to hens that did

not visit the range (indoor). The alarm call playback was proposed to elicit fear when it was played, but anxiety once it stopped. It was predicted the indoor birds would pay greater attention toward the surrounding environment manifested as greater vigilance behavior and delayed willingness to feed indicating higher anxiety than indoor birds. These indoor birds would also exhibit a stress-induced hyperthermic response with a drop in peripheral temperatures. The second objective was to determine an appropriate dose rate of the anxiogenic drug *m*-CPP for young-adult laying hens using an open field test. This drug was predicted to induce anxiety in chickens resulting in reduced locomotion but more vocalizations. The third objective was to compare the responses of birds given *m*-CPP or a saline solution in an attention bias test to pharmacologically validate this test as a measure of anxiety for use in welfare assessment of domestic poultry. It was predicted that the *m*-CPP dosed birds would show greater vigilance, reduced locomotion, more vocalizations and a longer latency to feed.

# MATERIALS AND METHODS EXPERIMENT 1: INDOOR AND OUTDOOR FREE-RANGE HENS

## Animals and housing

Research was approved by the CSIRO FD McMaster Laboratory Chiswick Animal Ethics Committee (ARA 12-13) in compliance with the Australian Code for the Care and Use of Animals for Scientific Purposes (*National Health and Medical Research Council, 2004*).

A total of 200 ISA Brown pullets were obtained from a commercial supplier in 2012 at 18 weeks of age and housed at CSIRO's Chiswick Poultry Facility, Armidale, NSW. The birds had been floor-reared until 8 weeks of age, then housed in cages until 18 weeks of age. Birds were housed indoors in an experimental tunnel-ventilated shed with straw litter area of 0.5 m$^2$/hen, access to water nipples, suspended pan feeders, nest boxes, and perches. All of the provided resources per bird exceeded the current Model Code of Practice for Domestic Poultry (*Primary Industries Standing Committee, 2002*). The birds were fed commercial layer pellets ad libitum. Lighting duration in the shed gradually increased up to a schedule of 15 h L:9 h D. Starting at 22 weeks of age, birds were given daily access to a range area of 12.5 m$^2$ per bird or 800 birds/hectare (at maximum occupancy) from 09:30 to 16:00 (pop-holes required manual operation). Range access was available via four pop-hole passageways (36 cm H × 18 cm W). The range area was grassed but devoid of trees, shelter or shade structures. All birds were visually inspected daily.

All birds were leg-banded upon arrival with numbered adjustable bands (Roxan Developments Ltd, Selkirk, Scotland) containing a Trovan® Unique ID 100 (FDX-A) microchip: operating frequency 128 kHz. Individual range use was tracked for 51 days from first pop-hole opening using RFID technology. The RFID systems were designed and supported by Microchips Australia Pty Ltd (Keysborough, VIC, USA) with equipment developed and built by Dorset Identification B.V. (Aalten, Netherlands) using Trovan® technology. Antennas situated within the pop-hole passageways registered the date, time and microchip of the individual bird as it went out onto the range, and again when the bird came back inside. The functionality of all microchips were confirmed prior to

placement on birds and at the end of the experiment. Further details on the specifications and validation of the RFID system are available in *Campbell et al. (2017)*.

## Experimental protocols

At 30 weeks of age, 36 hens were selected based on their range use patterns. "Outdoor" birds were defined as birds that ranged daily with 20 hens selected that also comparatively ranged for the longest period of time daily, and 16 "indoor" birds that were never registered to go outside (this was the total sample size of indoor birds). These birds were then tested in a manual restraint test, tonic immobility test, and an open field test (carried out in the same arena as the attention bias test) as part of a separate dataset (CE Hernandez et al., 2014, unpublished data). Each behavioral test was conducted once per bird. At 33–35 weeks of age the birds were also trained for tests of judgement bias as part of a separate dataset which involved several weeks of handling (CE Hernandez et al., 2014, unpublished data). The birds had daily range access on days when tests did not occur, but range use was not further tracked past 30 weeks of age when the indoor/outdoor birds were identified. At 51 weeks of age, birds were tested in an attention bias test. Range use may have changed during this period but other studies carried out with RFID tracking indicate outdoor birds are consistent in daily access across time (*Campbell et al., 2017*) and birds that remained indoors for the first 8 weeks of range access are distinct from birds that ventured outside immediately.

## Attention bias test

The arena used for the attention bias test was a wooden square open field arena of 1.25 m L × 1.22 m H and elevated 0.24 m off the ground with a clear frontal Perspex sliding panel. This arena was located in a separate enclosed room adjacent to the main poultry shed. Birds that were not being tested (i.e., within their home pens) could not hear the alarm call being played. A small pile of mixed grain was present in the back center of the arena. Birds were randomly selected from the home pen, held for thermal images (see following section for further details), carried to the test arena, placed in the front center and the test began. A conspecific alarm call was played immediately from portable speakers (MODEL #42820 ALUK0037708; Altec Lansing Technologies, InMotion, Milford, PA, USA). Alarm calls had been previously opportunistically recorded using a different group of 100 ISA Brown hens that were disturbed by a wild bird making noise outside the shed. As the alarm calls were produced by hens, they were likely signaling a "ground predator" (*Wilson & Evans, 2012*) which should elicit a "vigilant, erect posture" (*Evans, Evans & Marler, 1993*). Recordings were made using a Roland R-05 MP3 recording device (Roland Corporation US, Los Angeles, CA, USA) with a sampling rate of 48 kHz, and a bit rate of 128 kbps. The audio clips were edited into short playback files using the software Audacity 2.0.3 (http://audacity.sourceforge.net/). The same alarm call (8 s length) was played for all birds at a volume perceived to be similar to an alarm call volume of hens but precise details on the dB are unavailable. A video camera recorded all bird behavior and a second camera was connected to a screen to allow monitoring of the bird by an experimenter that remained out of sight. Once the bird reached the grain

pile, it was permitted 5 s to consume grain, then the alarm call was played a second time to gain a measure of their latency to resume feeding. Once the bird resumed eating the grains the test concluded, with a maximum test time of 10 min allotted.

## Behavior analysis

All videos were coded by a single experimenter using The Observer XT 12.0 (Noldus Information Technology, Wageningen, The Netherlands) who was blind to the ranging status of each hen and trained in the behavioral observations. Behavior was quantified to provide latency to first step, first vocalize, first eat, and the latency to eat following the second alarm call. Vigilance behavior was observed for the first 30 s following the first alarm call. Vigilance was quantified as the bird visibly stretching their neck while looking around, turning of the head may/may not also occur. Definitions of vigilance vary between laying hen studies (*Newberry, Estevez & Keeling, 2001*; *Odén et al., 2005*; *Zidar & Løvlie, 2012*) with this definition chosen as these clearly definable behaviors were observed. The terms of this behavior were agreed upon by three different observers prior to video observations but subsequently coded by a single observer only. Behavior was also observed during the alarm call to quantify if birds were responding to the playback as an indication that they were perceiving it as a threat. A "yes" response was categorized as exhibiting vigilance, a "maybe" response was categorized as the bird standing still but moving their head to look around (without neck stretching) and a "no" was quantified as the bird walking or eating with no apparent alteration in behavior during the playback. Intra-observer reliability for vigilance behavior across eight birds was 91% as assessed by correlation analysis in Microsoft Excel. Intra-observer reliability for latency measures across eight birds was 95–97%. A second observer watched the behavior during the alarm call to confirm 100% agreement with the assigned categories.

## Peripheral temperature assessment

To measure bird's temperature responses to the alarm call, a FLIR® ThermaCAM S60 was used (with emissivity set at 0.98) to take thermal images of the bird's head approximately every 10 s for the first 2 min of the test when possible. Because the test concluded following the second time eating, the total test length varied between birds but the first 2 mins included the majority of tested birds. The camera was placed within a cut-out on the side of the open field arena. To obtain baseline temperature values, images were first taken in the home pen (bird was unrestrained), and again with the bird held by the experimenter to allow clear images to be obtained. The bird was placed in the attention bias arena following collection of these images, which occurred up to approximately 2 mins before the test began. Ambient temperature (°C) and relative humidity (%) inside the sampling area were recorded every 30 min and entered into the infrared camera to calibrate it for atmospheric conditions. The maximum temperatures of the eye, comb, and head (as defined in *Edgar et al., 2013*) were recorded from the thermal images using image analysis software (ThermaCam Researcher 2.7; FLIR Systems AB, Danderyd, Sweden) and averaged per bird across the baseline images, then 1 min periods for the first 2 mins of the test. Hens that finished the test before 2 mins and hens that did not have clear

images at each time point were excluded ($n = 12$ outdoor, $n = 12$ indoor birds remained for analyses).

## Data and statistical analyses

All data were checked for homoscedasticity of the residuals to proceed with parametric tests. The time spent vigilant was analyzed using a two-tailed $t$-test to compare range use groups. The latencies to first step, vocalize and eat (after first and second alarm call) were analyzed using separate Kaplan–Meier estimates with a log-rank test for differences between range use groups. Birds which failed to perform each behavior within 600 s were censored results as per methods for adjusting survival data to account for individuals which did not reach a specific point within the allotted timeframe (*Leung, Elashoff & Afifi, 1997*). A total of 18 outdoor and 15 indoor birds were included in the analysis of the response to the second alarm call. The average max temperatures for the bird's head, comb and eye from baseline up to 2 mins post alarm call were compared using a General Linear Mixed Model with range use groups (indoor/outdoor) and time (baseline, 0–1 and 1–2 min post alarm call) as fixed effects including their interaction, bird ID was included as a random effect, nested within range use group. Restricted maximum likelihood estimation methods were applied. The interactions were not significant (all $P \geq 0.29$) and thus were removed from the final model. Post hoc students $t$-tests were conducted on the least squares means. All analyses were conducted in JMP 13.0.0 (SAS Institute, Cary, NC, USA) with $\alpha$ set at 0.05.

## RESULTS EXPERIMENT 1: INDOOR AND OUTDOOR FREE-RANGE HENS

The majority of birds showed clear vigilant responses to the alarm call playback with the exception of two birds that were categorized as "maybe" showing a response, and two birds were categorized as a "no" response. Two outdoor and one indoor bird never received a second alarm call and reached the maximum test time.

The indoor birds spent more time vigilant in the first 30 s following the first alarm call than the outdoor birds ($t = 4.74$, d$f = 41.20$, $P < 0.001$; mean ± SEM outdoor birds: 11.78 ± 1.82 s, indoor birds: 24.56 ± 1.99 s). The outdoor birds were faster to eat following the second alarm call than the indoor birds ($\chi^2 = 6.76$, d$f = 1$, $P = 0.009$, Fig. 1). However, there were no differences between range use groups in latency to eat following the first alarm call ($\chi^2 = 0.60$, d$f = 1$, $P = 0.44$; mean ± SEM outdoor birds: 176.23 ± 33.80 s, indoor birds: 227.54 ± 34.50 s), the latency to first step ($\chi^2 = 0.05$, d$f = 1$, $P = 0.82$; mean ± SEM outdoor birds: 141.37 ± 26.98 s, indoor birds: 151.56 ± 30.20 s), or the latency to first vocalize ($\chi^2 = 0.93$, d$f = 1$, $P = 0.33$; mean ± SEM outdoor birds: 49.16 ± 13.13 s, indoor birds: 62.21 ± 11.28 s).

There was a significant difference across time for the comb temperature ($F_{(2,46)} = 12.39$, $P < 0.001$) with the highest temperature at baseline (mean ± SEM baseline: 32.59 ± 0.58 °C, 0–1 min: 31.13 ± 0.58 °C, 1–2 min: 31.43 ± 0.58 °C). The indoor birds also showed significantly lower comb temperatures overall ($F_{(1,22)} = 4.79$, $P = 0.04$; mean ± SEM indoor birds: 30.50 ± 0.48 °C, outdoor birds: 32.93 ± 0.48 °C). There were no significant
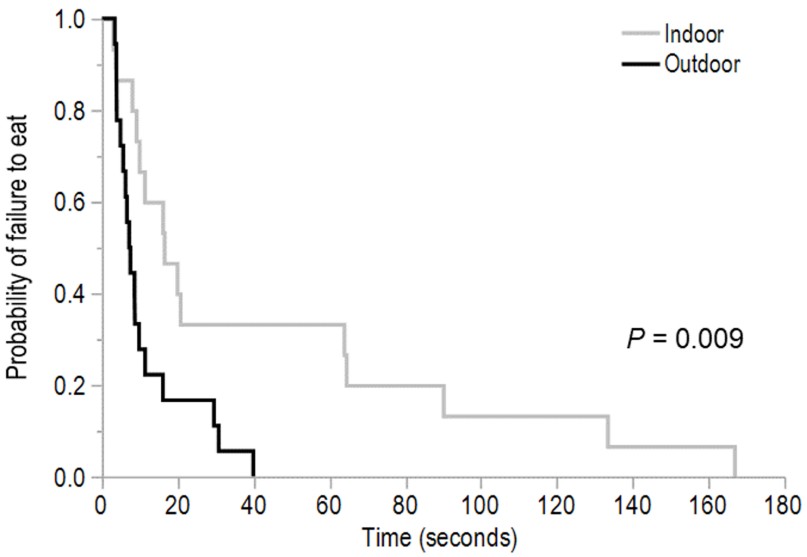

**Figure 1 The time to first eat following the second alarm call.** This Kaplan–Meier curve shows the latency in seconds to first eat for daily ranging (outdoor) and non-ranging (indoor) hens following a second alarm call played in the attention bias test. The plot indicates the probability that a bird would not eat at a given time point for the duration of testing.

differences across time for the head ($F_{(2,46)} = 3.09$, $P = 0.06$; mean ± SEM baseline: 38.21 ± 0.11, 0–1 min: 37.93 ± 0.13, 1–2 min: 38.06 ± 0.13), or eye ($F_{(2,46)} = 0.71$, $P = 0.50$; mean ± SEM baseline: 36.62 ± 0.17, 0–1 min: 36.36 ± 0.11, 1–2 min: 36.60 ± 0.13) temperatures nor differences between range use groups in the overall head ($F_{(2,46)} = 3.11$, $P = 0.09$; mean ± SEM outdoor birds: 38.24 ± 0.09, indoor birds: 37.89 ± 0.10), or eye ($F_{(1,22)} = 0.78$, $P = 0.39$; mean ± SEM outdoor birds: 36.65 ± 0.12, indoor birds: 36.47 ± 0.11) temperatures.

## MATERIALS AND METHODS EXPERIMENT 2: DRUG DOSE DETERMINATION

### Animals and housing

All research was approved by the University of New England Animal Ethics Committee (AEC 15-129) in compliance with the Australian Code for the Care and Use of Animals for Scientific Purposes (*National Health and Medical Research Council, 2004*).

Commercial floor-reared ISA Brown pullets ($n = 102$) were obtained at 15 weeks of age in 2017 and housed at the University of New England's Laureldale Poultry Facility. All birds were leg-banded upon arrival and equally divided between two adjacent floor pens (4.8 m L × 3.6 m W) at an approximate stocking density of three birds per m². Each pen contained perches, nest boxes, a single round feeder, a water nipple line, and rice hulls as floor litter. All of the provided resources per bird exceeded the current Model Code of Practice (*Primary Industries Standing Committee, 2002*). The birds were fed ad libitum a pullet grower feed, followed by a commercial layer mash from 18 weeks of age. The fluorescent lights in the shed were on a 16 h L:8 h D schedule with fan ventilation but no temperature or humidity control. Birds were visually inspected daily. A third adjacent

pen of identical resources and configuration was available to temporarily house birds during testing periods. There were no visual barriers between the adjacent pens.

## Drug treatments

Four different treatments were used to determine the optimal dose rate of $m$-CPP. These included a saline solution (control), 0.5, 1, and 2 mg/kg of $m$-CPP mixed into a saline solution at a dosing volume of 1:1. These drug dosages were selected based on prior studies using $m$-CPP with rats and sheep (*Bilkei-Gorzó, Gyertyán & Lévay, 1998*; *Lee et al., 2016*). The drug was mixed a maximum of 24 h prior to use and kept refrigerated (4 °C) while not in use. A total of 60 hens were randomly allocated to the four treatment groups ($n = 15$ birds/treatment). The majority of the birds ($n = 51$) came from within a single pen, with remaining birds taken from the second pen ($n = 9$) and distributed across the treatment groups. We acknowledge that birds from separate home pens may behave differently but we determined the impact of home pen would be minimal as the birds were from the same rearing flock and the home pens were directly adjacent allowing both visual and acoustic contact between neighboring birds. There were also no direct statistical comparisons between groups of birds from the separate pens (i.e., between Experiment 2 and Experiment 3) and where applicable, birds from the two home pens were balanced across a statistical treatment comparison. Birds were individually weighed (Bat1; VEIT Electronics, Moravany, Czech Republic) 1 day prior to the first day of testing to provide body weights for calculating individual dosages.

The open field test was used to behaviorally assess the impacts of the drug as this test is known to measure fear and desire for social reinstatement in chickens (*Forkman et al., 2007*); social reinstatement has been used as an indicator of anxiety (*De Haas et al., 2014b*). On the three testing days, individual birds, 19 weeks of age, were captured in their home pen and carried to a table located within the same shed. The bird was placed on its side with two handlers securing the bird gently (including covering their head) while a third person administered a subcutaneous injection of the drug or saline at the shoulder site. After each bird was injected, it was placed into the temporary holding pen. The first bird of the day was in visual contact with the neighboring pen of birds but physically isolated. Subsequent birds were not physically isolated as both newly dosed and tested birds were placed into the same pen to either wait for the drug to take effect or to observe for any longer-term drug effects across the day. A 20-min waiting interval for the drug to take effect was determined based on rat studies with $m$-CPP (*Bilkei-Gorzó, Gyertyán & Lévay, 1998*). Previous drug studies with young chicks have applied a 15-min waiting interval (*Sufka et al., 2009*; *Hymel & Sufka, 2012*) but there has been no previous application of $m$-CPP on adult hens. The same waiting interval was applied to birds injected with saline. During this time, experimenters were able to watch the birds from a distance to ensure no observable adverse effects of the drug. On the first day of testing, the first hen received a dose of $m$-CPP at 0.5 mg/kg, followed by a dose of 1 mg/kg on the second hen, and a dose of 2 mg/kg on the third hen. Almost no visible effects were observed with the hens in their pen (and specifically no observable adverse effects) and thus the drug treatment doses henceforth were administered in a random order.

## Open field testing

Individual birds were re-caught from the temporary pen just prior to the conclusion of the 20-min interval following drug administration and carried to the open field test arena located in a separate room. At exactly 20 min following dosing, the bird was placed into the front center of the open field arena with the room lights turned off. The wooden open field arena was a square of 1.25 m L × 1.22 m H and elevated 0.24 m off the ground with a clear frontal Perspex sliding panel (the same as per used in Experiment 1). Lights were then switched on and an open field test began. One video camera recorded all bird behavior for later analysis and a second camera was connected to a screen so all birds could be visually monitored during the test by an experimenter who remained within the room but out of sight. The open field test was 10 min in duration. At the conclusion of the test the lights were turned off and the bird was carried back to the temporary holding pen to allow further visual monitoring for any adverse effects of the drug.

## Video observations

The videos were watched by a single observer blind to the dosages for each hen who was first trained in all types of behavioral observations that were made. The observer recorded the latency to first move (when the hen moved more than just their head but did not take a step), latency to first step, and latency to first vocalize. Measuring the latency to first move differentiated the birds that were motionless in position upon placement in the arena vs those that were moving their body but cautious to step. The number of steps and number of vocalizations were also recorded within 1-min intervals. Any individual noise made by the hen was recorded as a single vocalization. The vocalizations were also re-classified by a separate trained observer (blind to treatments) into comparatively "short" and "long" calls. No bird made alarm calls while in the arena. A random selection of 12 videos were looked at by a second observer with the inter-observer reliability correlation at 95–98% across all variables as assessed by correlation analysis in Microsoft Excel.

## Data and statistical analyses

All latency data (first move, step, and vocalize) and counts of steps and vocalizations per 1 min intervals were collated for each dosing treatment group. The latency to first move, step, and vocalize data were analyzed using Kaplan–Meier estimates with a log-rank test for differences between treatment groups. Birds which failed to perform each behavior within 600 s were censored results. The count data were square-root transformed and analyzed using General Linear Mixed Models that included the effects of treatment, time, and their interaction. Non-significant interactions were removed from the final model. Bird ID nested within treatment was included as a random effect. Restricted maximum likelihood estimation methods were applied. All analyses were conducted in JMP 13.0.0 (SAS Institute, Cary, NC, USA) with $\alpha$ set at 0.05.

## RESULTS EXPERIMENT 2: DRUG DOSE DETERMINATION

The Kaplan–Meier estimates indicated an effect of drug dosing treatment in the latency to first move ($\chi^2 = 15.67$, d$f = 3$, $P = 0.001$, Fig. 2), but no differences between dosing

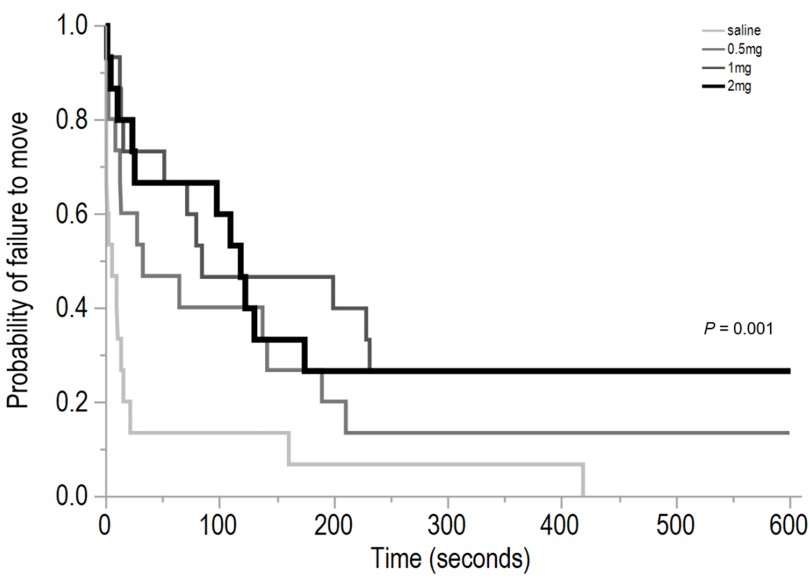

**Figure 2 Time to first move for hens from four dosing groups.** The Kaplan–Meier curve shows the latency for hens that were dosed with saline (control), 0.5, 1, or 2 mg/kg of *m*-CPP) to first move (seconds) in an open field test. This plot indicates the probability that a hen would not move at a given time point for the duration of testing.

treatment groups in the latency to first vocalize ($\chi^2 = 6.02$, d$f$ = 3, $P = 0.11$; mean ± SEM saline: 114 ± 54, 0.5 mg/kg: 156 ± 61, 1 mg/kg: 147 ± 53, 2 mg/kg: 317 ± 71), or the latency to first step ($\chi^2 = 3.27$, $P = 0.35$; mean ± SEM saline: 211 ± 66, 0.5 mg/kg: 271 ± 63, 1 mg/kg: 335 ± 62, 2 mg/kg: 355 ± 66).

There was no overall effect of dosing treatment on the number of steps birds made across the 10 min test duration ($F_{(3,56)} = 2.09$, $P = 0.11$; mean ± SEM saline: 8.41 ± 0.97, 0.5 mg/kg: 6.4 ± 0.90, 1 mg/kg: 4.41 ± 0.70, 2 mg/kg: 1.82 ± 0.40). However, visually the data were distinct and thus more focused post hoc tests were applied (*Hsu, 1996*) which did show significantly fewer steps were made by the 2 mg/kg dosed birds compared to the saline dosed birds. There was also an effect of time ($F_{(9,531)} = 5.11$, $P < 0.001$) with comparatively fewer steps made during the first 2 min of the test (Table 1). There was no interaction between treatment and time ($P = 0.63$) and this was removed from the final model. When step data from the first 2 min of the test only were analyzed, there was a significant effect of dosing treatment ($F_{(3,56)} = 3.34$, $P = 0.03$) with birds given both 1 and 2 mg/kg *m*-CPP stepping less than birds given saline (Fig. 3). There was also an effect of time ($F_{(1,59)} = 5.55$, $P = 0.02$) with fewer steps made in the first minute of the test (Table 1) but no interaction between time and dosing treatment group ($P = 0.29$) and this was removed from the final model. There was an interaction between dosing treatment and time in the total number of vocalizations made ($F_{(27,504)} = 1.64$, $P = 0.02$) and an overall effect of time ($F_{(9,504)} = 7.93$, $P < 0.001$) with more vocalizations made in the first 4 min of the test (Table 1). There was no overall effect of dosing treatment group on total vocalization number ($F_{(3,56)} = 1.0$, $P = 0.40$; mean ± SEM saline: 3.69 ± 0.56, 0.5 mg/kg: 8.4 ± 1.62, 1 mg/kg: 3.73 ± 0.64, 2 mg/kg: 6.96 ± 2.0). When comparisons were made specifically on the number of short vocalizations made there was no effect of

**Table 1** Mean ± s.e.m of the number of steps or vocalizations (total, short, long) made by hens from four treatment groups across a 10 min open field test.

| | | 0–1 min | 1–2 min | 2–3 min | 3–4 min | 4–5 min | 5–6 min | 6–7 min | 7–8 min | 8–9 min | 9–10 min |
|---|---|---|---|---|---|---|---|---|---|---|---|
| All dosing treatments[1] | # steps | 2.01 ± 0.60[c] | 3.72 ± 0.98[b,c] | 5.47 ± 1.32[a,b] | 6.32 ± 1.34[a] | 7.1 ± 1.64[a] | 5.85 ± 1.33[a,b] | 5.83 ± 1.53[a,b] | 6.1 ± 1.38[a] | 5.15 ± 1.03[a,b] | 5.15 ± 1.13[a,b] |
| | # total vocals | 7.25 ± 1.57[a,b] | 8.27 ± 1.90[a,b] | 10.78 ± 3.35[a] | 7.92 ± 1.94[a,b] | 3.53 ± 0.93[b,c] | 2.75 ± 1.0[c] | 1.95 ± 0.96[c] | 5.38 ± 2.60[c] | 5.65 ± 2.48[c] | 3.47 ± 3.05[c] |
| | # short vocals | 6.82 ± 1.55[a,b] | 7.72 ± 1.89[a,b,c] | 10.48 ± 3.35[a] | 7.52 ± 1.94[a,b] | 3.3 ± 0.89[b,c,d] | 2.67 ± 1.0[d] | 1.88 ± 0.96[d] | 5.37 ± 2.60[c,d] | 3.6 ± 1.60[d] | 3.38 ± 3.05[d] |
| | # long vocals | 0.47 ± 0.14[a] | 0.48 ± 0.19[a,b] | 0.3 ± 0.11[a,b,c] | 0.13 ± 0.06[b,c,d] | 0.23 ± 0.10[a,b,c,d] | 0.08 ± 0.04[b,c,d] | 0.07 ± 0.05[c,d] | 0.02 ± 0.02[d] | 0.07 ± 0.05[c,d] | 0.03 ± 0.03[d] |

**Notes:**
[1] Values are presented for all dosing groups combined (saline, 0.5, 1, 2 mg/kg).
[a,b,c,d] Different superscripts within rows indicate significant differences between times as determined using post hoc analyses ($P < 0.005$).

treatment ($F_{(3,56)} = 1.18$, $P = 0.33$; mean ± SEM saline: 3.18 ± 0.52, 0.5 mg/kg: 8.19 ± 1.62, 1 mg/kg: 3.57 ± 0.63, 2 mg/kg: 6.15 ± 1.87), but more were made during the first 4 min of the test ($F_{(9,531)} = 7.55$, $P < 0.001$, Table 1) and there was no interaction between treatment and time ($P = 0.10$) so this was removed from the final model. There was a significant interaction between treatment and time for the long vocalizations ($F_{(27,504)} = 2.16$, $P < 0.001$) with saline birds making the most long vocalizations during the first 2 mins of the test (Table 1). Overall, the 2 mg/kg birds made fewer long vocalizations than the saline and 0.5 mg/kg birds ($F_{(3,56)} = 3.03$, $P = 0.03$) but the numbers of these calls were low (mean ± SEM saline: 0.36 ± 0.09, 0.5 mg/kg: 0.23 ± 0.06, 1 mg/kg: 0.14 ± 0.05, 2 mg/kg: 0.03 ± 0.03) and reduced significantly across time ($F_{(9,504)} = 4.95$, $P < 0.001$, Table 1).

# MATERIALS AND METHODS EXPERIMENT 3: PHARMACOLOGICAL VALIDATION OF AN ATTENTION BIAS TEST

## Habituation phase

All research was approved by the University of New England Animal Ethics Committee (AEC 15-129) in compliance with the Australian Code for the Care and Use of Animals for Scientific Purposes (*National Health and Medical Research Council, 2004*).

A total of 50 birds were used to pharmacologically validate an attention bias test. This sample size included the remaining birds that were not previously used for pharmacological dosing trials in Experiment 2 and also 14 birds that had previously been given only a saline dose in the open field test (Experiment 2). These previously-tested birds were subsequently divided evenly between treatment groups for the attention bias test. However, to reduce the chance of birds responding to the novelty of the test arena with freezing behavior (as observed in Experiment 2) rather than responding to the alarm call playback, all birds were habituated to the open field arena at 21–22 weeks of age. This was to closer approximate the tests done by *Brilot & Bateson (2012)* where an alarm call was played to birds in their home cages. For habituation, all birds were caught from their home pen and individually placed into the front-center of the open field arena with the lights off. Lights in the room were turned on and each bird was given 10 min to become familiarized with the environment. Lights were turned off after 10 min and the bird was carried back to the home pen (an additional leg band identified habituated

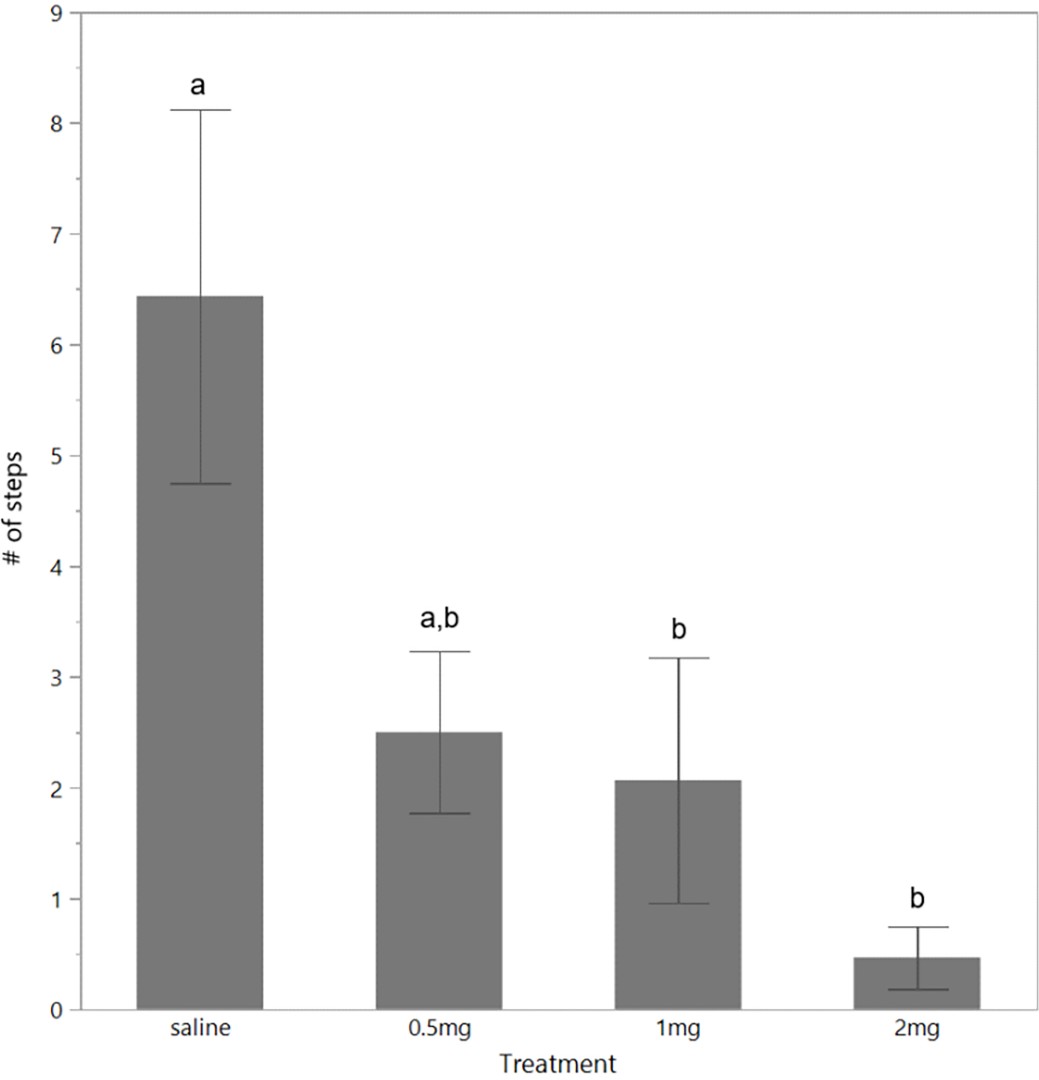

**Figure 3 Steps made in the open field test.** The mean (± SEM) number of steps made in the first 2 mins of the open field test by hens dosed with saline (control), 0.5, 1, or 2 mg/kg of *m*-CPP. Different superscript letters indicate significant differences between treatment groups at $P < 0.05$.

birds to avoid recapture). This was repeated three times for each bird across a period of 6 days. In these habituation trials, a small pile of mixed poultry grain was placed center-back of the arena. Birds had been exposed to the grain scattered in their home pens across three days prior to habituation and were observed to consume the grain. The 14 saline birds that had been included from the drug dose trials only had two habituation sessions with the grain, their first time in the open field arena during the previous trials was considered their first habituation. As per Experiment 2, a video camera recorded all bird behavior, and an out-of-sight experimenter watched live via a second camera connected to a screen to ensure birds were becoming habituated to the test arena. Birds were also feed-restricted the evening prior to habituation session 3. A reduced amount of food was available within the home pen compared to previous ad

libitum supply so that birds were limited in the amount of food they could consume but they were not food-deprived. All birds were observed to rapidly consume the grain by the third habituation session and thus were deemed ready to be tested in the attention bias test.

### Attention bias test

At 23 weeks of age, birds ($n = 50$) were randomly allocated to the 2 mg/kg $m$-CPP treatment or saline (control) group and individually weighed to calculate their dosage volume. The drug was mixed into a saline solution at a dosing volume of 1:0.5 to reduce the volume of liquid given to the birds. Birds were feed-restricted in the home pen the evening prior to testing to increase motivation to eat, but were not food-deprived. Birds were then tested in the attention bias arena at 23 weeks. Four birds were excluded due to unexpected external disturbance outside the facility during testing (remaining test birds: saline $n = 24$; 2 mg/kg, $n = 22$).

The $m$-CPP or saline was administered sub-cutaneously as outlined in Experiment 2 and birds were placed into the temporary pen. At 20 min following administration, birds were placed front-center of the open field arena in the dark. The lights were then switched on and an attention bias test began. An alarm call was played back for 8 s immediately following lights-on at 75–80 dB (as measured inside the box by a Digitech QM-1589 digital sound level meter; DigiTech, Sandy, UT, USA). The alarm calls were played from Latitude E7270 laptop speakers located on one side of the open field arena. Two different alarm calls were randomly-allocated between birds but the same alarm call was played up to twice for each bird. Alarm calls were selected from the same set of recordings made from a group of hens prior to Experiment 1 but different calls were used compared to the one used in Experiment 1. Following the first alarm call, an out-of-sight observer watched on the connected screen until the bird reached the feed and consumed grain for 10 s, then the alarm call was played a second time for 8 s. This second alarm call was played to gain a measure of whether the alarm call was sufficiently threatening to cause the bird to stop feeding, and also to measure the bird's willingness to resume feeding again. If a bird did not approach and consume the feed, no second alarm call was played as the second measure was aimed at measuring their response once they had started eating. All tests were video-recorded for later decoding of behavior. All tests concluded after 5 min and the bird was carried back to the temporary pen. All birds were tested once across 2 days, and placed back in their home pen at the conclusion of testing.

### Video observations

The third habituation session videos were watched by a single experimenter to record latency to first step, vocalize, and eat. The Observer XT 12.0 (Noldus Information Technology, Wageningen, The Netherlands) was used by a single trained experimenter who was blind to the treatments of each hen to decode all attention bias videos. The experimenter measured latency to first step, first vocalize, and first eat as well as latency to eat, step, and vocalize following the second alarm call. The total number of steps and individual vocalizations were counted across the test duration. Finally the total time spent

feeding and duration of individual feeding bouts (following the second alarm call only) were recorded where a single feeding bout was defined as the bird eating the food without raising its head or walking away from the feed. Intra-observer reliability correlations between a selection of 10 videos observed twice were 93–96% as assessed by correlation analysis in Microsoft Excel. Clear vigilance behavior with stretching of the neck as measured in Experiment 1 was not observed in this different set of birds and thus was not assessed further.

## Data and statistical analyses

To determine if the birds were perceiving the alarm call playback as a threat, the latency to first step, vocalize, and eat were compared between habituation session 3 and the attention bias test using Kaplan–Meier estimates with a log-rank test for differences between tests separately for saline and 2 mg/kg hens. The same Kaplan–Meier estimates were also used to compare the latency (seconds) to first step, vocalize, and eat between the two treatment groups in the attention bias test. Hens which failed to perform each behavior within 600 s were censored results. The hens that never received a second alarm call were included in latency data analyses but excluded from all subsequent analyses (saline: $n = 2$; 2 mg/kg: $n = 5$, i.e., a total of 22 saline and 17 2 mg/kg birds were included). The total number of steps and total number of vocalizations were converted to a rate of steps or vocalizations made per second of the test, excluding all time spent eating for the two treatment groups. These data were logit transformed to approach normality and analyzed using a two-tailed $t$-test. The total time (seconds) spent eating and duration of eating bouts (following the second alarm call) were square-root transformed and analyzed using a two-tailed $t$-test. Where applicable, the untransformed values are presented in the results as there was no difference between the raw and back-transformed means. All analyses were conducted in JMP 13.0.0 (SAS Institute, Cary, NC, USA) with α set at 0.05.

## RESULTS EXPERIMENT 3: PHARMACOLOGICAL VALIDATION OF AN ATTENTION BIAS TEST

The Kaplan–Meier estimates showed a significantly slower latency to eat ($\chi^2 = 6.99$, df = 1, $P = 0.008$) during the attention bias test compared to habituation session 3 for the saline birds, but a significantly faster latency to vocalize ($\chi^2 = 16.77$, df = 1, $P < 0.001$) and no difference in the latency to step ($\chi^2 = 1.67$, df = 1, $P = 0.20$, Figs. 4A–4C). The 2 mg/kg hens also showed significantly slower latencies to step ($\chi^2 = 6.88$, df = 1, $P = 0.008$) and eat ($\chi^2 = 20.75$, df = 1, $P < 0.001$) during the attention bias test compared to habituation session 3, but faster latencies to vocalize ($\chi^2 = 19.40$, df = 1, $P < 0.001$, Figs. 4D–4F).

There was a significant effect of dosing group for the latency to first eat during the attention bias test following the first alarm call ($\chi^2 = 5.65$, df = 1, $P = 0.018$) with the 2 mg/kg hens taking longer to eat (Fig. 5A). Similarly, the 2 mg/kg birds also took longer to eat following the second alarm call ($\chi^2 = 11.23$, df = 1, $P < 0.0008$, Fig. 5B). There was no effect of dosing treatment group in the latency to first step ($\chi^2 = 1.61$, df = 1,

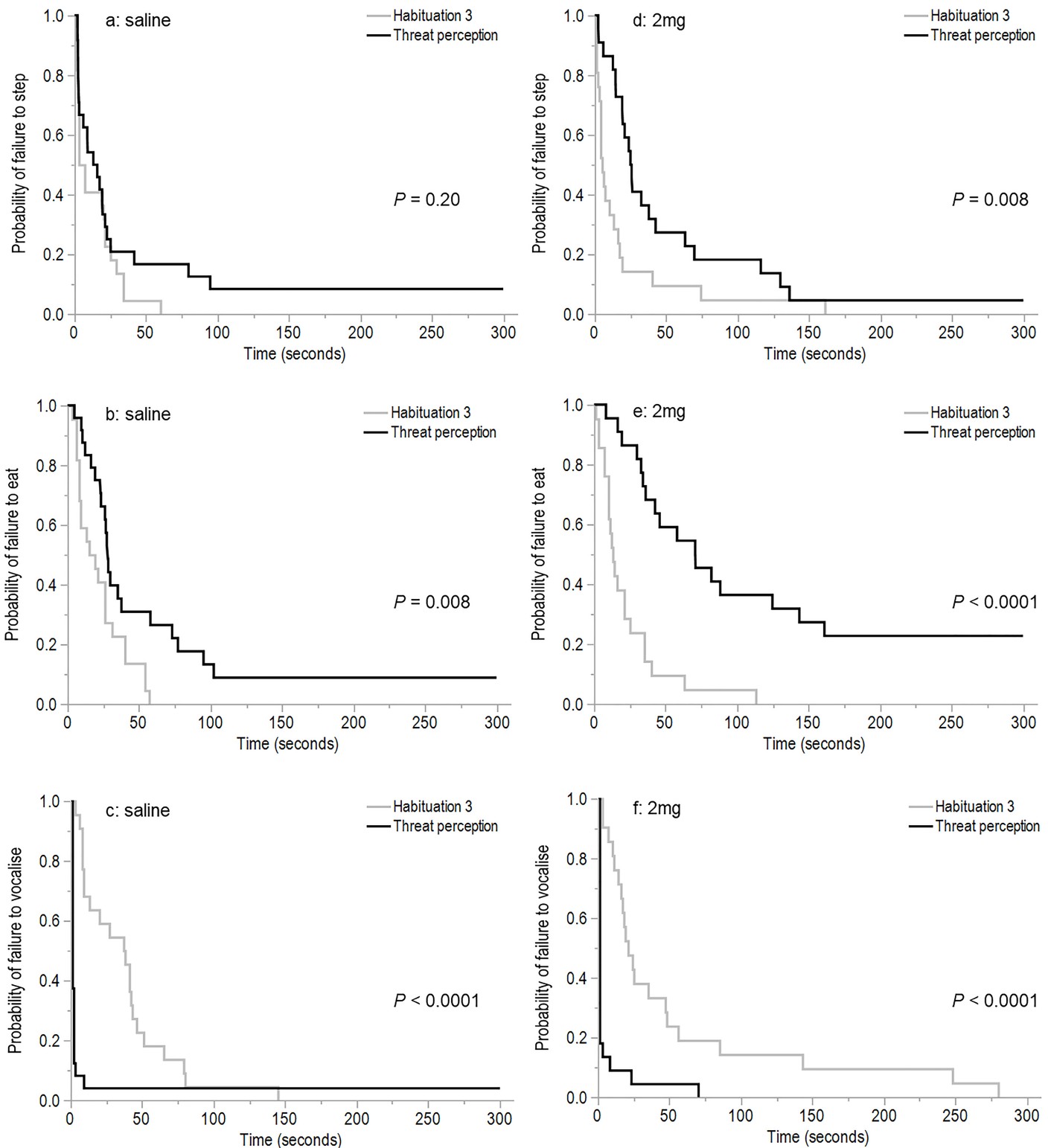

**Figure 4** **The time to first step, eat, and vocalize for hens dosed with saline or *m*-CPP.** The Kaplan–Meier curves show the latency (seconds) for hens to first step (A), eat (B), and vocalize (C) during habituation session 3 and the attention bias test when dosed with saline solution during the attention bias test only, and the latencies to first step (D), eat (E), and vocalize (F) during habituation session 3 and the threat perception test for hens when dosed with 2 mg/kg *m*-CPP during the threat perception test only.           

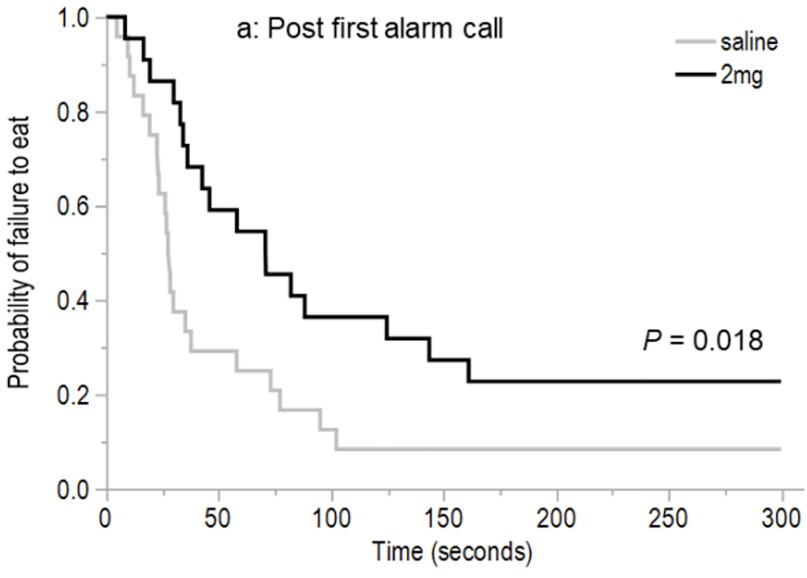

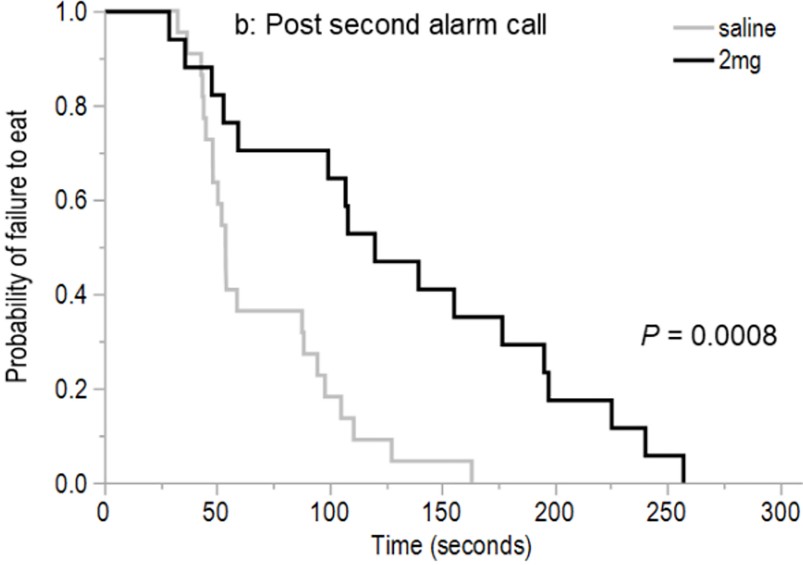

**Figure 5 The time to eat following the first and second alarm calls.** The Kaplan–Meier curves show the latency to eat (seconds) following the first alarm call (A) and latency to eat (seconds) following the second alarm call (B) for hens dosed with 2 mg/kg *m*-CPP or a saline solution in the attention bias test. Two saline-dosed and five 2 mg/kg-dosed birds did not eat and thus never received a second alarm call playback (*n* = 7 birds not included in graph "B").

$P = 0.20$; mean ± SEM 2 mg/kg: 51.89 ± 14.51 s; saline: 42.32 ± 16.89 s) and the latency to first vocalize following the first alarm call ($\chi^2 = 0.05$, df = 1, $P = 0.82$; mean ± SEM 2 mg/kg: 78.42 ± 23.40 s, saline: 110.21 ± 28.18 s) or the latency to first step following the second alarm call ($\chi^2 = 0.06$, df = 1, $P = 0.81$; mean ± SEM 2 mg/kg: 109.0 ± 16.24 s, saline: 115.31 ± 15.49 s). However, there was a significant effect of dosing treatment group on the latency to vocalize ($\chi^2 = 4.83$, df = 1, $P = 0.03$) following the

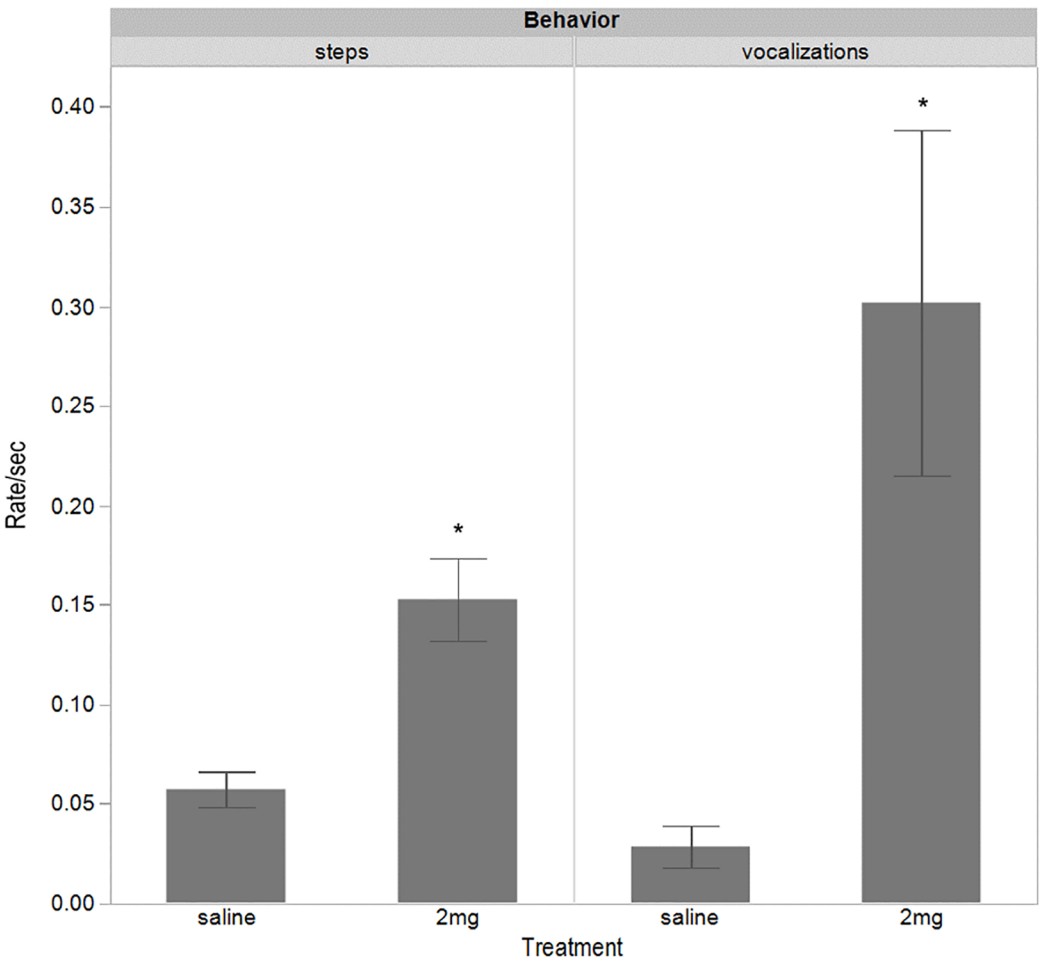

**Figure 6 Proportional steps and vocalizations for each second of the attention bias test when not eating.** The data are displayed for hens dosed with 2 mg/kg of *m*-CPP or a saline solution. Asterisks indicate significant differences (*P* < 0.001) between dosing treatment groups for each measured behavior.

second alarm call with the 2 mg/kg hens vocalizing sooner (mean ± SEM 2 mg/kg: 118.84 ± 15.84 s, saline: 175.31 ± 16.77 s).

The 2 mg/kg birds spent less time eating than the saline hens ($t = 3.69$, d$f = 43.93$, $P = 0.0003$; mean ± SEM 2 mg/kg: 12.65 ± 4.12 s, saline: 31.0 ± 3.62 s) but there was no difference between dosing treatment groups in the average eating bout length ($t = 0.46$, d$f = 93.80$, $P = 0.65$; mean ± SEM 2 mg/kg: 14.06 ± 2.0 s, saline: 14.64 ± 1.22 s).

There were significant effects of dosing treatment on the rate (steps or vocals per test time excluding eating) of stepping ($t = -4.63$, d$f = 38.08$, $P < 0.001$) and vocalizing during the test ($t = -3.65$, d$f = 22.75$, $P = 0.001$) with the 2 mg/kg birds stepping and vocalizing more (Fig. 6).

## DISCUSSION

An attention bias test for chickens could be used as an indicator of anxiety. In laying hens, an attention bias toward the surrounding environment following playback of an alarm call

simulating a threat, may be indicated by increased vigilance, reduced locomotion, and a reduced willingness to feed, but behaviors of test birds may be impacted by their age and/or familiarity with the testing environment. The attention bias test behaviorally distinguished indoor-preferring and outdoor-preferring free-range hens at 51 weeks of age where indoor-preferring hens showed a perceived heightened awareness of the simulated threat as indicated by increased vigilance and a reduced willingness to eat. The drug *m*-CPP affected behavior of 19-week-old adult layers at a dose of 2 mg/kg which manifested as reduced locomotion in an open field arena. Hens habituated to the open field arena and given *m*-CPP at 23 weeks of age showed increased locomotion and vocalizations and reduced willingness to eat in the attention bias test but no evidence of the same vigilance behavior as observed in the first experiment with free-range hens.

Based on previous studies of behavioral assessment in free-range hens under controlled settings, indoor hens display higher levels of fear than outdoor ranging hens (*Campbell et al., 2016*; *Hartcher et al., 2016*; *Hernandez et al., 2014*). The free-range environment is unpredictable with exposure to predators. Thus birds that remain indoors may choose to do so because of fear. But often these indoor birds are never registered to visit outdoors and thus their choice to stay inside is likely due to a perceived threat which can be classified as anxiety. As predicted, the indoor birds assessed in an attention bias test were more vigilant following the alarm call and slower to start feeding. Both of these behaviors combined suggest increased anxiety but no formal assessment was made of any potential differences in hunger states between the two groups of birds. There was a drop in peripheral temperature from baseline to indicate stress-induced hyperthermia (*Herborn et al., 2015*), and the indoor birds also had lower comb temperatures overall. Chickens have previously shown a decrease in peripheral temperature as a result of air-puffs and handling stress (*Edgar et al., 2011*, *2013*; *Herborn et al., 2015*) validating this as a measure of distress, although *Moe et al. (2012)* did demonstrate that arousal in anticipation of a reward also resulted in a decrease in peripheral temperature. *Edgar et al. (2013)* documented a temperature drop 1 min following handling but as the free-range birds tested in the current study were accustomed to regular handling, the temperature decrease is interpreted to be in response to the alarm call. This temperature drop suggests higher arousal in the indoor birds, but the valence (positive/negative) of the response is equivocal. Additional physiological measures could be made in future testing scenarios and that may confirm valence. Catheter blood-sampling could further characterize the physiological stress responses such as production of plasma catecholamines, or corticosterone (*Korte et al., 1997*; *Larsen et al., 2018*). However, caution would need to be placed on the interpretation of physiological measures that could be affected by both a state of anxiety and activity.

Dosing of *m*-CPP in adult laying hens affected their behavior in the open field arena. Anecdotally, in the home pen, birds were observed to walk cautiously/slowly around the pen and several birds exhibited an odd behavior of picking up feathers/litter off the ground and trying to place it on their backs. It is difficult to prove or dis-prove any hallucinogenic effects of the drug in hens, but it is a potential side-effect to be considered (*Tancer & Johanson, 2001*). These odd behaviors were not seen in saline-dosed birds.

However, we do not know if there was any social transmission of negative affective states or stress between tested/untested or saline/*m*-CPP dosed birds kept within the same housing pen prior to testing (*De Haas et al., 2012*). Purposeless oral movements as a side effect of *m*-CPP have been reported in rodents (*Kreiss & De Deurwaerdère, 2017*) and humans can suffer from migraines followed administration of *m*-CPP (*Panconesi, 2008*). To the best of the author's knowledge, the pharmacokinetics of *m*-CPP in laying hens has not previously been documented or the presence of side effects which may not be overtly displayed. In humans, the bioavailability and half-life varies substantially between subjects, which may only partly be explained by genetic variation in the metabolizing enzymes (*Feuchtl et al., 2004*). The cytochrome P450 enzymes are responsible for the metabolism of *m*-CPP (*Rotzinger et al., 1998*) and are present in chickens (*Watanabe et al., 2013*) but research to document the pharmacokinetic pathways is still needed. Further pharmacological validation of behavioral tests with layers using alternative anxiogenic drugs with different modes of action such as caffeine or FG-7142, for example, would be valuable.

The open field arena was selected as an environment to clearly assess behavior at the individual-bird level. At higher doses of *m*-CPP, the birds in this arena exhibited prolonged freezing and reduced locomotion. Freezing in an open field arena is a typical response of laying hens experiencing an actual threat (visible human threat: *Suarez & Gallup, 1982*), as well as birds in the arena that have not specifically been presented with a threat (*Gallup & Suarez, 1980*). In this study, *m*-CPP accentuated the freezing response for birds placed in the open field arena for the first time with no specific known threat presented. The open field test instigates a trade-off in chickens between the desire for social reinstatement and a reduction in locomotion/freezing to avoid detection (*Gallup & Suarez, 1980*). The vocalizations emitted at the different doses of *m*-CPP indicated that the type of vocalization made by the bird could be more informative vs total number of vocalizations, but further detailed spectrographic analysis would be required which was beyond the scope of the current study. Additional dosing trials using *m*-CPP could document the behavioral responses of birds both when the environment is novel, and then when the environment is familiar.

In comparison to the final habituation session, the playback of the alarm call during the attention bias test reduced the willingness to feed in both saline and *m*-CPP birds indicating the birds were perceiving the alarm call as threatening. However, the birds in Experiment 3 did not exhibit the same vigilance behavior (stretching of the neck and looking around) as did the birds in the first experiment when comparing between indoor and outdoor free-range hens. This could be the result of a multitude of differences between the testing groups such as age, breed, housing environments, or prior testing experience. Early life experience with cognitive stimulation has been demonstrated to affect displays of vigilance as adults (*Zidar et al., 2017*). Vigilance is also defined differently across laying hen experiments (*Newberry, Estevez & Keeling, 2001*; *Odén et al., 2005*) and thus one definition may not fit all testing scenarios. Further refinement and use of an attention bias test is needed to develop a set of standard behavioral measures. Although the birds in Experiment 3 did not exhibit vigilance similar to the free-range hens,

the *m*-CPP birds were less willing to eat which suggests an increased attention bias. Both sheep and cattle dosed with *m*-CPP also showed reduced willingness to feed during an attention bias test (*Lee et al., 2016*, *2018*; *Monk et al., 2018a*). The *m*-CPP drug is, however, also an appetite suppressant. Humans report a reduced appetite while dosed with *m*-CPP (*Thomas et al., 2014*), but in one study it only affected consumption of a palatable snack when human subjects were already satiated (*Thomas et al., 2018*). Feeding behavior alone in this study is likely not sufficient to distinguish between the dosing groups, but in combination with the increased locomotion and vocalizations, the *m*-CPP birds could be interpreted as being more anxious following the playback of the alarm calls. Similar to this study, in other tests with livestock, the *m*-CPP dosed individuals also exhibited other changes supplement to feed-related behavior such as paying more attention to the location where the dog was presented, more vigilance, and increased head and tail-related behaviors in cattle (*Lee et al., 2018*). In further refinements of an attention bias test for laying hens, a different reward could be used that would avoid any confounds of pharmacological side-effects on appetite. A practical test developed to be potentially used on-farm for sheep successfully used an image of a conspecific rather than feed (*Monk et al., 2018b*). Willingness to forage in new substrate for example, could be an alternative to a food reward for hens.

In contrast to the freezing behavior that was observed with dosed birds placed into the open field arena for the first time, 2 mg/kg *m*-CPP birds previously habituated to the open field arena showed increased locomotion and vocalizing relative to control birds. In sheep, *m*-CPP did not affect locomotion or vocalizations in comparison to control animals (*Lee et al., 2016*; *Monk et al., 2018a*), but number of zones crossed increased for cattle dosed with *m*-CPP (*Lee et al., 2018*) and in rats, higher doses of *m*-CPP reduced (*Kennett & Curzon, 1988*; *Kennett et al., 1989*) or tended to reduce locomotion (*Kreiss & De Deurwaerdère, 2017*). If freezing is an indication of high anxiety, we would have predicted the playback of an alarm call would have increased anxiety and resulted in reduced locomotion as observed in the open field test. Given the contrast in responses between non-habituated and habituated birds (of the same breed/flock and similar age), the change in locomotion is unlikely to be an effect of the drug per se, but an indication of how familiarity of the environment impacts the responses the birds exhibit. When hens were more familiar with the test environment, the anxiety manifested as increased movement, potentially the hens were looking for escape from the arena and reinstatement with their peers. Pacing behavior has been observed in birds believed to be trying to escape during behavioral testing (*Bolhuis et al., 2009*), and exhibited by hens frustrated by the non-delivery of an expected food reward (*Zimmerman & Koene, 1998*). Although the hens during the test were not observed to be stereotypically pacing in the arena, there are possibly similar underlying affective states that led to increased movement in the current study. These results suggest that extreme behavioral responses (freezing/high locomotion) in an open field test may both be indicative of negative affective states. Further comparisons with more groups of birds would be highly valuable as individual groups of birds in the same housing environments can display different behavioral patterns (*Campbell, Horton & Hinch, 2018*).

## CONCLUSIONS

Overall, an attention bias test could be used with laying hens to measure anxiety but it requires further refinement. The familiarity of the birds with the testing environment will likely affect their behavioral responses where habituation to the testing environment may result in improved detection of anxiety specifically in response to the alarm call playback. Subsequent iterations of the test may result in a set of standardized behavioral measures to be used. The drug $m$-CPP appears to be most effective at a dose rate of 2 mg/kg for pharmacological validation of this affective state in laying hens. Measuring anxiety will provide understanding of how hens may be responding to their housing environments and the impacts it may have on their welfare. Extreme behavioral responses of freezing or high locomotion exhibited in individual-bird behavioral tests are possibly both indicators of negative states.

## ACKNOWLEDGEMENTS

Thank you to Deborah Rufo (Agro-Paris Tech), Georgia Ballard (University of New England), and Tim Dyall (CSIRO) for assistance with data collection. Thank you to Andrew Cohen-Barnhouse (University of New England) for the drug dosage calculations. Thank you to Jim Lea (CSIRO) for the behavioral analysis of the video recordings and general management of the birds.

### Funding

Funding for this project was provided by the University of New England (School of Environmental and Rural Science project expense support) and the Poultry CRC, projects 1.5.2, 1.5.6. There was no additional external funding received for this study. The funders had no role in study design, data collection and analysis, decision to publish, or preparation of the manuscript.

### Grant Disclosures

The following grant information was disclosed by the authors:
University of New England (School of Environmental and Rural Science project expense support) and the Poultry CRC, projects 1.5.2, 1.5.6.

### Competing Interests

The authors declare that they have no competing interests.

### Author Contributions

- Dana L.M. Campbell conceived and designed the experiments, performed the experiments, analyzed the data, prepared figures and/or tables, authored or reviewed drafts of the paper, approved the final draft.
- Peta S. Taylor performed the experiments, authored or reviewed drafts of the paper, approved the final draft.

- Carlos E. Hernandez conceived and designed the experiments, performed the experiments, analyzed the data, approved the final draft.
- Mairi Stewart performed the experiments, contributed reagents/materials/analysis tools, approved the final draft.
- Sue Belson performed the experiments, contributed reagents/materials/analysis tools, approved the final draft.
- Caroline Lee conceived and designed the experiments, performed the experiments, contributed reagents/materials/analysis tools, authored or reviewed drafts of the paper, approved the final draft.

## Animal Ethics

The following information was supplied relating to ethical approvals (i.e., approving body and any reference numbers):

Experiment 1 was approved by the CSIRO FD McMaster Laboratory Chiswick Animal Ethics Committee (Animal Research Authority 12–13).

Experiments 2 and 3 were approved by the University of New England Animal Ethics Committee (Animal Ethics Committee 15–129).

## Data Availability

The raw data for these studies are available at: Campbell, Dana; Lee, Caroline; Belson, Sue; Lea, Jim (2018): Laying hen attention bias test data. v3. CSIRO. Data Collection. DOI 10.25919/5cbd1ca76aa19.

## Supplemental Information

Supplemental information for this article can be found online at http://dx.doi.org/10.7717/peerj.7303#supplemental-information.

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
