# Peer review of "An attention bias test to assess anxiety states in laying hens"

_PeerJ, doi:10.7717/peerj.7303_

## Round 0.1 · original submission · Major Revisions

As you can see, the two Reviewers differ somewhat in their comments. Please take into account all of their quite detailed comments (with particular attention to comments on vocalizations by Reviewer 2).
I look forward to seeing a revised version of your paper.

Best regards,
Giorgio Vallortigara

Reviewer 1 ·

Basic reporting

In terms of sufficient field background provided, it is suggested that the authors include more on peripheral temperature assessment as an indicator of animal affect state in the introduction, such as physiological mechanisms behind temperature increase/decrease given context and perception of animal.

Line 103-105: Perhaps include a semicolon between "chicks" and "interpreted" so make this sentence more clear, or, rewrite entire sentence.

Line 115: Can the authors define "low-ranging birds"? Is that based on distance from pop-hole, or, active movement throughout day, or, duration of day spent outside?

Line 454-461: There appears to be a repetition of the sentences. Can these be distinguished better, or, remove the redundancy?

Line 415: Confusing sentence, could it be rewritten? For example, "Based on previous studies of personality assessment in free-ranging hens, indoor hens display higher levels of fear trait than outdoor free-range hens."

Figure 6: Asterisk is used to indicate significant difference, but in Figure 3, superscript letters are used. Could the authors use a consistent format (unless this is a journal preference).

Experimental design

Line 172: What is the operational definition used to identify "outdoor birds"? Was this a minimum of 4 hours spent on range, or, a minimum of 20% of access time spent on range (for example).
Line 199-200: Did the attention bias testing occur in the same building as the home pens? Could the birds hear the alarm call when not in the test arena? If so, how did the authors account for this confound?
For each experiment, the inter-observer reliability was based on 20% of the data, correct? If this is sufficient cross-coding for reliability assessment, please provide a reference as justification.
Line 309-310: The major confound of birds in the study coming from two separate home pens was not taken into consideration during analysis. Could the authors provide evidence of no impact of home pen on behavior in experiments (or at least attempt to justify this confound)?
Line 326-327: While it is assumed, it is not clearly stated that saline injected birds also waited 20 minutes in the holding pen. In addition, the rationale for using rat model of cMPP metabolism/effect time is not given. It is assumed that no other species (avian, for example) has been examined for correct dosage or time for effect?
Line 390-392: This inclusion of 14 previously handled and isolated saline birds does not appear to be justified other than to increase sample size. These birds experienced the stress of handling and the pain of saline injection, therefore, they could be biased in their interpretation of the alarm call. Also, were these birds evenly split between the two home pens?
Line 418: How feed restricted were the birds? Based on body weight, or reduction in feed quantity?

Validity of the findings

Line 261: It is more appropriate to say that "The majority of birds showed clear vigilant behavior..." instead of "All", since it is stated that 7 birds did not show clear vigilance.
Line 372-373: The authors should remove this statement of non-significant post hoc test (fewer steps by 2 mg/kg dosed birds) as it is misleading.
Line 375-378: No justification is given for why the authors looked to the first 2 minutes of the 10 minute open field test, other than the assumption that the authors are manipulating the data in order to show significance. Is there something about poultry and immediacy effects of novelty or isolation? Why are the first 2 minutes of isolation important?
Line 504: The authors mention that the age of the bird could impact behavior with no evidence to back this claim.
line 538: While this reviewer does not support anecdotal reports in scientific discussions, if this is supported by the editor, than the potential hallucinogenic effects of mCPP should be discussed as well.

Additional comments

This is an exceptionally designed study of using attention bias testing to infer negative affective state in chickens. The authors created a logical story bringing three superb experiments together to tell one concise story. The "proof of concept" objective in the use of anxiogenic drug validated future use of attention bias to assess husbandry and management practices which could impact the welfare state (i.e., result in chronic anxiety) for commercial birds. Hypotheses for each study were clearly written, and the difficulties of assessment (such as, definition of vigilance) was well-written. The limitations of this study are also given in a clear and honest fashion, such as the appetite suppressant effects of mCPP. This is a robust study with clear implications for future comparative cognitive studies of poultry affective states, as well as a tool to assess animal welfare of birds under varying housing/management conditions.

Reviewer 2 ·

Basic reporting

I think the manuscript has clear English used throughout. The literature cited is biased towards domestic references, although references within the topic in a slightly more behavioural ecology setting are available and would complement the current references (mainly regarding personality, and behaviour of fowl). I think the structure of the paper is clear. Raw data is not shared what I can see. Results are presented following the structure of the outlined hypotheses.

I think the manuscript “fails” on these aspects (presented in no specific order):
-There are several types of attention biases that can take place, hence the authors need to make this clearer, and also not call their test “THE attention bias test”, but “AN attention bias test”, or more honestly, “responses to playback of alarm call”.

-Regarding the claimed novelty of the use of responses to playback of alarm call. I find this not really appropriate since such a playback call has been used before in behavioural research in domestic fowl (see e.g. Favati et al 2014 Plos One, which seems relevant to this work).

-The authors have currently not described what type of vocalisations they recorded (e.g. in the variables analysed, such as latency to vocalise, number of vocalisations). Chickens can produce very many different calls with different suggested motivation (contact calls, distress calls). The authors have this information in their videos and this could clear out some confusion about the not clear responses of birds with regards to vocalisation. Also, chickens have 2 very distinctive alarm calls for different predators, and this is not clearly explained in the manuscript, nor clearly presented what was used in the study. Further, I suggest to look over and make sure it is clear why 2 alarm calls were played back to each bird, and also then what type of alarm calls. Also, it is not clear to me why a 2nd alarm call was not played if the bird did not resume eating after the first call (L433).

-I think there are too many variables presented. This is of course not a problem per see, since many aspects of behaviour were indeed recorded. My problem is more that several of the measured variables are very likely to be strongly correlated (either positively, like latency to move, latency to first step – and what are the exact biological meanings of differences in these? Or negatively, like vigilance and eating/moving). By reducing the variables that are strongly correlated will both reduce the number of results presented (which the authors currently do not make specific predictions or interpretations of, and should increase the readability of their work) and also claimed “double findings” where in fact only 1 result exists (e.g. feeding vs vigilance if strongly correlated, as they typically are in these kind of tests in chickens). Further, it is not clear to me (from the information the authors provide) how the open field is a social reinstatement test, and which response variable that capture this, in the current study. Overall are each behaviour measured not clearly justified and motivated why chosen, and how it is predicted to be affected by the various treatments.

-The use of different groups of birds make it hard to interpret results across experiments (e.g. vigilance, as defined was not observed in experiment 3). The authors do however try to be clear about this restriction of their work.

-The authors refer to ‘personality’ (which is not defined in the ms as far as I can see) and also papers claimed to look at this in the fowl. I only went through Campbell et al 2016, which is a reference for this, and that paper did not test repeatability of behaviour, hence has not investigated personality. What I suggest that the authors do, is just to rephrase (define personality if using it, not call behaviour personality if not demonstrated repeatability/consistency etc), and also to refer to some of all the literature that has described personality in the fowl (work by Favati et al, Zidar et al).

-Why temperature was measured, is not clearly set up or motivated, in the current methods/manuscript.

Experimental design

The manuscript presents work that fit within the scope of the journal. The research question is relatively well defined. It is not fully clear how the presented research fills a knowledge gap, but I believe the authors will be able to erect this. The investigation is relatively rigorous and with high technical and ethical standards (although weaker with regards to what behaviours were recorded and why). Methods are mainly described with sufficient details.

-The largest problem with the experimental design is the use of different groups across testing, which makes interpretation across the 3 experiments, hard.

Validity of the findings

The situation where it is harder to interpret the overall results because experiments were carried out on different groups of birds, is presented by the authors. There are controls within each experiment, and data has ok to low sample size (down to n=9 I think). I am a bit puzzled by the choice of non-parametric analyses of data that so clearly is not normally distributed (e.g. latencies, or categorical data with 3 categories only), but the authors have checked assumptions for the use of parametric statistics, so it should be ok. Conclusion stated are overall unclear due to the design of the study, but linked to the original research question. But from the suggested need to clear up co-varying response variables presented as separate results, I think it should be ok. I think it is clear when the authors are speculating (but from the above regarding personality).

I think the manuscript “fails” on these aspects (presented in no specific order):

-Please provide analysed sample sizes for experiment 1 and 3.

- When recording facial temperature, is body temperature not important to also record and relate to?

- I think the authors need to look over that they do provide descriptive statistics (mean +- SE or equivalent) throughout. E.g. “Result Experiment 3” lacks this (and lack test statistics for several analyses).

-There are more potential responses to a potential threat than only freeze. In general, freeze, fight or flight.

Additional comments

I enjoyed reading your work, and I think the aim to find a relatively simple test to measure fear/anxiety in domestic animals, is a great aim within animal welfare. It is nice to see that the authors aim to validate the test pharmacological.

Of other comments that the authors may want to take part of, I have these:

- I think the style of talking about ‘A’ and not ‘THE’ test is also valid for e.g. Open field test, as there are many variations on this test too. But this is a personal preference of mine, and most open field tests aim to measure relatively similar things/behaviour.

-L150: it is unclear to me what floor reared and cage-reared exactly mean.

-Experimental protocols, around L175: Is the use of the same room later used for playback of alarm call, also used for relatively negative tests (e.g. manual restrain test etc) a problem due to negative associations with the room?

-I believe Zidar & Lovlie 2012, Favati et al 2014 (Plos One), Zidar et al 2018 (BES) describe vigilance in this way in the fowl.

-L240: By excluding birds that did not finish the test within only 2 min, I think you remove relevant behavioural variation.

-Statistical analyses throughout. The authors keep repeating that they “censored the results” for birds that got max values. It is said in the exact same phrasing, and never explained. Please consider explaining why, and what consequences this has.

- For the description of the statistical analyses of the first experiment, all fixed effects are not defined (i.e. groups used = , time = ).

- There is quite a lot of overlap regarding information on housing and animal housing across the experiments, that I think the authors can reduce.

-The heading “open field testing” is one paragraph too early for experiment 2, in my mind.

- L418-19: Look over these sentences, as the feed restriction and not food deprived information is not clear to me.

- Is not recorded number over time, rate and not proportion?

-Are there problems with differences observed in already the first open field of hens exposed to 2mg/kg vs. controls, considering they both should be habituated to the arena?

-Again semantics, there is no “alarm call threat” (L502) but a playback of an alarm call, which we think should function as a threat. Further, we do not know that hens show an increased perception of the alarm call, but we assume things around the playback (L506). Please re-phrase.

-Zidar et al 2017 (Behav Proc) show that early life experiences affect vigilance in the fowl.

-L520-536: It seems to me that the temperature response to playback is similarly hard to interpret as other tests then, if both excitement and anxiety/fear can result in a drop in temperature (similar to how increase in cort can be both due to stressed and active individuals).

-I do not follow why the authors think that the use of a different reward during playback would be helping in better understand what the response of a bird means (L591-). Please develop.

---

## Round 0.2 · Minor Revisions

Both Reviewers are happy with your revision, and requested only some minor further revision.

If you address these changes and resubmit, I will be happy to accept the paper.

Reviewer 1 ·

Basic reporting

No comment.

Experimental design

No comment.

Validity of the findings

No comment.

Additional comments

The authors' response to the reviewer's recommendations were sufficient and changes made to the manuscript are acceptable. No additional changes are recommended by this reviewer.

Reviewer 2 ·

Basic reporting

This is the 2nd time I review this ms, and I am pleased with the revision the authors have produced. I only have minor comments on the current version (see below).

Experimental design

No further comments.

Validity of the findings

No further comments.

Additional comments

My additional minor comments on your work are:
L136-: A reference more relevant to your work, is comparison of personality and coping style in fowl (Zidar et al 2017 Animal Behaviour), more than comparison with social status?
L285-: All the 3 result sections. I think it is more transparant to present Mean and SE also for non-significant result, than to only due so for signitficant ones.
L361 & 365: I would change the word significant to 'observable' here.
L365: I would clarify what the random aspect refers to.
L386: Consider re-phrasing 'frozen' since the bird did not litteraly freeze.
L408-: This section lacks test-statistcs L417, 422, 426, 428, although interactions (so both for consistency and clarity I would present test-statistics for also these).

---

## Round 0.3 · accepted · Accept

It seems that you have satisfactorily replied to all reviewers' concerns. I am thus happy to accept the paper in the present form.